# Distributed Consensus Kalman Filter Design with Dual Energy-Saving Strategy: Event-Triggered Schedule and Topological Transformation

**DOI:** 10.3390/s23063261

**Published:** 2023-03-20

**Authors:** Chunxi Yang, Gengen Li, Guanbin Gao, Qinghua Shi

**Affiliations:** 1Faculty of Mechanical and Electrical Engineering, Kunming University of Science and Technology, Kunming 650500, China; ycx@kmust.edu.cn (C.Y.); lg15571195216@163.com (G.L.);; 2Yunnan International Joint Laboratory of Intelligent Control and Application of Advanced Equipment, Kunming 650500, China; 3Yunnan Institute, China Academy of Machinery Science and Technology Group Co., Ltd., Kunming 650031, China

**Keywords:** event-triggered schedule, topological transformation, timeliness window, distributed Kalman filter

## Abstract

In the distributed information fusion of wireless sensor networks (WSNs), the filtering accuracy is commonly negatively correlated with energy consumption. Therefore, a class of distributed consensus Kalman filters was designed to balance the contradiction between them in this paper. Firstly, an event-triggered schedule was designed based on historical data within a timeliness window. Furthermore, considering the relationship between energy consumption and communication distance, a topological transformation schedule with energy-saving is proposed. The energy-saving distributed consensus Kalman filter with a dual event-driven (or event-triggered) strategy is proposed by combining the above two schedules. The sufficient condition of stability for the filter is given by the second Lyapunov stability theory. Finally, the effectiveness of the proposed filter was verified by a simulation.

## 1. Introduction

In recent decades, wireless sensor networks (WSNs) have been widely researched and applied in many fields, such as self-calibration, surveillance, and target tracking [1,2], owing to their small size, high flexibility, multiple functions, low costs, and simple installation [3,4,5,6]. WSNs are part of a fully distributed network consisting of many wireless sensors [7]. One of the core issues of their application is how to design a distributed filter to precisely estimate the system state [8]. Considering the Kalman filter is one of the most classical linear filters and is successfully applied in many fields, many researchers have combined the filter with WSNs to extend their applications [9,10,11,12,13,14].

In contrast with wired sensor networks, although WSNs have many advantages (as mentioned above) there are still some limitations, including limited energy, poor bandwidth, short communication distance, weak computing, storage capabilities, etc. [15,16]. The research predicted that the CO${}_{2}$ emission would be over 1400% (1900 baseline at 100%) and that the primary energy consumption would be over 300% (1970 baseline at 100%) by 2022 [17]. Thus, from the view of energy conservation and emission reduction, as well as the extended lifetime of WSNs, it is necessary to design energy-saving strategies for WSNs. It is well known that an event-driven strategy is one of the most representative solutions to the problem of limited energy [18,19], it can effectively save energy by avoiding unnecessary information transmission [20]. As a result, many scholars have studied it from a communication perspective. The distributed estimation problem of a network sensing system with an event-driven schedule has been analyzed in [21]; the authors proposed the event-triggered Kalman consensus filter by minimizing the mean square error based on event-driven protocols. Moreover, data transferring and scheduling were studied [22], and the lifetime of the network has been extended by reducing the communication bandwidth and improving energy efficiency.

In addition to the perspective of communication, some scholars have also considered the event-driven strategy from a data features perspective. The literature [23] demonstrates a data packet processor based on an event-driven approach. To save energy effectively, only necessary packets are selected for transmission after ensuring the performance of the $H_{\infty }$ filter. The problem of event-triggered state estimation in a linear Gaussian system with an energy harvesting sensor is studied in [24]. Moreover, the event-triggered condition is designed based on the importance of data and available energy, and then the frequency of data transmission is adjusted accordingly. In [25], a new event-driven strategy is proposed where the upper and lower bounds of the event-triggered threshold are time-varying and automatically adjusted. Although these works can effectively save energy through event-triggered scheduling, the influence of WSN topology on energy consumption is ignored.

In WSNs, the communication distance between sensors is the most important factor for energy consumption [26]. Because these connections determine the WSN topologies, topology control is reviewed in [27,28], including its control method, evaluation standard, and some common issues. In [29], the authors proposed a distributed topology control algorithm, which optimizes the topology based on the real-time residual energy of nodes. Similar works can be found in [30,31,32,33,34,35].

However, a large number of research studies have ignored the fact that sensor nodes have limited data storage capacity. According to [26], transferring data consumes more energy than collecting it. To balance estimated accuracy and energy savings, this paper proposes a fully distributed state estimator with a dual energy-saving strategy (e.g., an event-triggered schedule and topological transformation).

In addition, the packet loss factor is essential when designing a WSN with good performance [36]. It is widely known that excessive packet loss can significantly impact the quality of information fusion. As sensors can only store a small amount of data in finite steps, the event-driven strategy designed in this paper effectively mitigates the effects of packet loss by utilizing historical data. In practice, these two strategies are interdependent and both impact the estimation performance of WSNs. Thus, the filter with dual energy-saving strategies can not only save energy but also promote more uniform energy consumption. Moreover, it can improve the robustness and extend the lifetime of WSNs by making full use of the node’s storage ability. The main contributions of the paper are summarized as follows:We propose a unique data fusion strategy (see Equation (Equation 9)), according to five communication situations between nodes *i* and *j*. It can effectively decline the effect of packet loss in the network in full using historical data.We propose a new topological transformation rule (see Equation (Equation 7)) based on the energy consumption model of nodes in WSNs. It avoids the single node from consuming energy too fast. Thus, the WSN lifetime is extended.We designed a novel distributed consensus Kalman filter based on an event-triggered schedule and topological transformation. Unlike the generally distributed Kalman filter, the proposed filter with a dual energy-saving strategy is able to offer more possibilities for energy savings in WSNs.

The remainder of this paper is organized as follows. In Section 2, we present some mathematical preliminaries required in this paper. In Section 3, we introduce the distributed estimation framework and design the energy-saving strategy. In Section 4, we formulate a type of distributed state estimator algorithm with dual driving and state the conditions for stability. Finally, we provide simulation verification of the proposed algorithm in Section 5; the conclusions are drawn in Section 6.

## 2. Mathematical Preliminaries

$R^{n\times m}$ denotes the set of real matrices with *n* rows and *m* columns. *I* is the *n*-dimensional identity matrix. $Z^{+}$ represents the set of positive integers. $diag(A_{1},\ldots ,A_{n})$ represents a block-diagonal matrix. In addition, E(·) represents the mathematical expectation.

Let G=(V,E;A) is an *m*-order undirected graph. $V=\{v_{1},\ldots ,v_{m}\}$ is a nonempty finite set of nodes and E⊆V×V is a set of edges. In addition, $\left(v_{i},v_{j}\right)$, i,j=1,2,…,m represents an edge of G. The weight adjacency matrix is denoted by $A=[a_{i,j}]$. $a_{i,j}\geq 0$ denotes the weight for the edge $(v_{i},v_{j})\in E$, which represents the closeness of the connection between any two sensor nodes. Meanwhile, we assume that $a_{i,i}=0$. The set of real-time neighbors of node *i* is denoted by $N_{RN}^{i}$.

In addition, to clarify the mathematical symbols used, we list them in Table 1, and the abbreviations used throughout the paper are listed in Table 2.

## 3. Problem Statement and Strategic Design

### 3.1. Problem Statement

To simplify the description, several assumptions are given (as follows).

**Assumption** **1.**
*Each sensor node is able to collect information from the monitored object, and it also receives the information from the neighbor nodes until the packet dropout happens.*


**Assumption** **2.**
*For all sensors, the state $x_{k}$ of the monitored object is the same at the k instance. This means $x_{k,i}=x_{k},i=1,2,\ldots ,m$ at the k instance, where m represents the total number of sensors in WSNs.*


If a WSN is deployed in the monitored area, the general frame of the event-driven distributed filter is shown in Figure 1.

Suppose *m* wireless sensors are randomly arranged in the monitored area. One of the main objectives is to obtain the required state estimate value by the distributed consensus Kalman filter with as less energy consumption as possible by means of the proposed event-triggered schedule and time-varying switching communication radius.

In general, the monitored object shown in Figure 1 can be described as a discrete linear time-invariant system as follows: (1)$$\left\{\begin{array}{c} x_{k+1}=Fx_{k}+Gw_{k}\\ y_{k,i}=Hx_{k}+v_{k,i} \end{array}\right.$$
where $k\in Z^{+}$ is the sampling instance, $x_{k}\in R^{n\times 1}$ is the state of the monitored object at the *k* instance, $y_{k,i}$ is the output of the *i*-th sensor node at the *k* instance. *F*, *G*, and *H* are constant matrices with compatible dimensions. $w_{k}$ and $v_{k,i}$ represent the system noise and measurement noise of the *i*-th sensor, respectively, at the *k* instance, which are assumed as unrelated Gaussian noises with zero means. Their covariance matrices are *Q* and $R_{k,i}$, respectively.

### 3.2. Energy-Saving Strategic Design

#### 3.2.1. Event-Triggered Schedule

Three new conceptions are defined below.

Timeliness window (TW): In WSNs, node *i* can store the information received from neighbor $j(j\in N_{RN}^{i})$ within a period of time and the information can be used by node *i* in this period. This period is defined as a timeliness window, also known as the timeliness period denoted by $\Delta (\Delta \in Z^{+})$. As a result, the neighbor node *j* in TW is called the timeliness neighbor of node *i* denoted by $j\in N_{TW}^{i}$.Real-time neighbor (RN): If node *j* sends information to node *i* at the *k* instance, then *j* is a real-time neighbor of node *i* at this sampling time. It is denoted by $j\in N_{RN}^{i}$.Effective neighbor (EN): If node *j* is the timeliness neighbor of node *i* or its real-time neighbor, then it is called an effective neighbor of node *i*. It is denoted by $j\in N_{EN}^{i}$. It is clear that $N_{EN}^{i}=N_{RN}^{i}+N_{TW}^{i}$.

The proposed event-driven principle is shown in Figure 2.

In Figure 2, node *i* receives the estimation ${\hat{x}}_{k,j}$ from its neighbor *j* and detects the output $y_{k,i}$ at the *k* instance. Then, the latest estimation ${\hat{x}}_{\tau ,j}$ in the buffer of node *i* is transformed to the local Kalman filter to obtain the estimation ${\hat{x}}_{k,i}$ at the *k* instance and send it to the event observer part and its real-time neighbor *j*. Finally, the event observer checks the even-driven condition (shown in Equation (Equation 2)) and determines whether to receive the next estimations from its neighbors or not.
(2)$$\delta_{k,i}={\left({\hat{x}}_{k,i}^{-}-F^{\tau_{k}^{i}}{\hat{x}}_{k-\tau_{k}^{i}}^{-}\right)}^{T}\left({\hat{x}}_{k,i}^{-}-F^{\tau_{k}^{i}}{\hat{x}}_{k-\tau_{k}^{i}}^{-}\right)>\delta$$
where ${\hat{x}}_{k,i}^{-}$ represents the prior estimation of node *i* at the *k* instance. $\tau_{k}^{i}\in \left\{0,1,2,\ldots ,\Delta \right\}$ is the difference between the last event-triggered time of node *i* and the current. ${\hat{x}}_{k-\tau_{k}^{i}}^{-}$ represents the prior estimation of node *i* at the latest event-triggered instance. In addition, δ is the event-triggered threshold.

If Equation (Equation 2) is satisfied, the event will be triggered. That means node *i* sends the command to node *j* ($j\in N_{RN}^{i}$), and node *j* will broadcast ${\hat{x}}_{k,j}^{-}$ to node *i*. Then node *i* will conduct consensus fusion based on the new information. Otherwise, node *i* will do it based on the latest information stored in its effective neighbor.

In WSNs, packet loss is a common phenomenon due to some unreliable factors, such as the time-varying bandwidth limitation, limited power, or uncertain environment. Considering the above event-triggered strategy, there are five communication situations between node *i* and *j*.
When the event is triggered ($\delta_{k,i}>\delta$), node *i* may receive the information coming from neighbor *j*.When the event is triggered ($\delta_{k,i}>\delta$), node *i* does not receive the information of neighbor *j* because the packet dropout occurred, but it stores the effective information of neighbor *j*.When the event is triggered ($\delta_{k,i}>\delta$), the information of neighbor *j* is continuously lost from the k−Δ instance to the *k* instance during the transmission, then the information of node *j* stored in node *i* is invalid.When the event is not triggered ($\delta_{k,i}\leq \delta$), the latest information of node *j* stored in node *i* is valid.When the event is not triggered ($\delta_{k,i}\leq \delta$), the information of *j* stored in *i* is invalid.

#### 3.2.2. Topology Transformation Schedule

To achieve energy savings by adjusting the topology structure of WSNs, an energy consumption model needs to be established. Figure 3 shows the well-known energy consumption model [26] of nodes in WSNs.

It is assumed that the sensor node does not consume energy during the measurement process. A sensor node transmits *l* bit of data with *d* m distance, the energy consumed by the sender and receiver are $E_{se}$ and $E_{re}$, respectively. They are calculated as follows [26].
(3)$$E_{re}\left(l\right)=l\cdot E_{elec}$$
(4)$$\begin{array}{cc} E_{se}\left(l,d\right) & =\left\{\begin{array}{c} l\cdot E_{elec}+l\cdot \varepsilon_{fs}\cdot d^{2}\;,\;\;d<d_{0}\;\\ l\cdot E_{elec}+l\cdot \varepsilon_{mp}\cdot d^{4}\;,\;\;d\geq d_{0}\; \end{array}\right. \end{array}$$
where $E_{elec}$ (nJ/bit) is the needed energy to send a 1-bit packet. $d_{0}$ (m) is the critical distance. $\varepsilon_{fs}$ (nJ/(bit$\cdot m^{2}$)) and $\varepsilon_{mp}$ (nJ/(bit$\cdot m^{4}$)) denote the energy consumption factors.

Including the computing energy consumption, the total energy consumption of node *i* in a data fusion process is $E_{i}^{to}$: (5)$$E_{i}^{to}=E_{se}\left(l,d\right)+E_{re}\left(l\right)+E_{ce}$$
where $E_{ce}$ (nJ/(bit · signal)) represents the energy for data aggregation in one period.

**Remark** **1.**
*It can be seen that $E_{se}\left(.\right)$ is proportional to $d^{2}$ or $d^{4}$ from the energy consumption model (4). Therefore, it is a good method to save energy by reducing the communication distance between nodes. However, to improve filtering accuracy, it is necessary to increase the communication distance of node i by increasing the number of its neighbors. Therefore, the selection of the communication distance between nodes is crucial to balance these two requirements.*


**Remark** **2.**
*The lifetime of WSNs is a critical factor to consider. Given that the node death can significantly reduce network connectivity, the lifetime of WSNs ultimately depends on the node that first runs out of energy.*


The communication radius $d_{i}$ of node *i* is defined as the maximum distance that it can transmit information. TO determine the communication radius switching rule, the local average energy of nodes *i* at the *k* instance is defined as $E_{k,i}^{av}$: (6)$$E_{k,i}^{av}=\frac{1}{N_{i}+1}\left(E_{k,i}+\sum_{j=1}^{N_{i}}E_{k,j}\right)$$
where $N_{i}$ represents the number of elements in $N_{RN}^{i}$; $E_{k,i}$ and $E_{k,j}$ denote the remaining energy of node *i* and *j*, respectively. So the switching rule for the communication radius of node *i* is given as follows.
(7)$$d_{k+1,i}=\left\{\begin{array}{c} d_{min},\;E_{k,i}>E_{k,i}^{av}\\ d_{max},\;others\;\;\;\;\;\; \end{array}\right.$$
where $d_{k+1,i}$ represents the communication radius of node *i* at the k+1 instance, $d_{min}$ is the minimum communication radius, and $d_{max}$ is the maximum. $E_{k,i}$ and $E_{k,i}^{av}$ represent the residual energy and the average residual energy of node *i* at the time *k* instance, respectively.

**Remark** **3.**
*In contrast to the fixed topology, Equation (Equation 7) can change the communication radius of node i according to the value of $E_{k,i}^{av}$. It avoids the single node from consuming energy too fast. Thus, the WSN lifetime is extended.*


## 4. Distributed State Estimator Design

Firstly, $x_{k,i}^{-}$$x_{k,i}^{+}\in R^{n\times 1}$ represent the prior estimate and posterior estimate of the system state for node *i* at the *k* instance, respectively. They are defined as follows.
(8)$${\hat{x}}_{k,i}^{-}=E\left(x_{k}|y_{1,i},y_{2,i},\ldots ,y_{k-1,i}\right),{\hat{x}}_{k,i}^{+}=E\left(x_{k}|y_{1,i},y_{2,i},\ldots ,y_{k,i}\right)$$

If the $\varphi_{k,i}=0$ represents packet loss and the $\varphi_{k,i}=1$ otherwise at the *k* instance, its value is controlled by the packet loss rate α. Suppose that the ${\hat{x}}_{k,j}$ is the latest neighbor information used by node *i* for consensus fusion. Then, the rules are given as follows.
(9)$${\hat{x}}_{k,j}^{-}=\left\{\begin{array}{c} x_{k,j}^{-},\;\delta_{k,j}>\delta \;and\;\varphi_{k,i}=1\\ x_{TW,j}^{-},\;\delta_{k,j}>\delta \;,\;\varphi_{k,i}=0\;and\;\tau_{k,j}\leq \Delta \\ x_{k,i}^{-},\;\delta_{k,j}>\delta ,\;\varphi_{k,i}=0\;and\;\tau_{k,j}>\Delta \\ x_{TW,j}^{-},\;\delta_{k,j}\leq \delta \;and\;\tau_{k,j}\leq \Delta \\ x_{k,i}^{-},\;\delta_{k,j}\leq \delta \;and\;\tau_{k,j}>\Delta \end{array}\right.$$
where $x_{TW,j}^{-}$ is the information of the timeliness neighbor of node *i*.

**Remark** **4.**
*After node i broadcasts its local information, the neighboring nodes may either receive the information or experience packet loss, which can be caused by unknown factors. Assuming packet loss is a uniformly random process, it can be considered a probabilistic event. Thus, the packet loss rate α (0≤α≤1) can be used to describe this phenomenon.*


Then, the current estimation ${\hat{x}}_{k,i}^{+}$ is given by
(10)$${\hat{x}}_{k,i}^{+}={\hat{x}}_{k,i}^{-}+K_{k,i}\left(y_{k,i}-H{\hat{x}}_{k,i}^{-}\right)+C_{k,i}\sum_{j\in N_{EN}^{i}}\left({\hat{x}}_{k,j}^{-}-{\hat{x}}_{k,i}^{-}\right)$$
where $K_{k,i}$ and $C_{k,i}$ represent the Kalman gain and consensus gain of node *i* at the *k* instance, respectively.

Next, the stability properties of the proposed algorithm are analyzed. For the reader’s convenience, all of the proofs are given in Appendix A.

**Theorem** **1.**
*Setting the consensus gain $C_{k,i}=0$ yields the sub-optimal Kalman gain, i.e., $K_{k,i}=P_{k,i}^{-}H^{T}{\left(HP_{k,i}^{-}H^{T}+R_{k,i}\right)}^{-1}$.*


**Theorem** **2.***The consensus Kalman filter is asymptotical stability if Equation (Equation 10) and $C_{k,i}=cP_{k,i}^{+}{\left(F-K_{k,i}HF\right)}^{-1}$ are used with the gain c satisfying the following condition.*(11)$$cI\leq \sqrt{\left[{\left(P_{k-1}^{+}\right)}^{-1}-L_{k}^{T}{\left(P_{k}^{+}\right)}^{-1}L_{k}\right]{\left[\Psi_{k}^{T}L_{k}^{-1}P_{k}^{+}{\left(L_{k}^{-1}\right)}^{T}\Psi_{k}\right]}^{-1}}$$
where $L_{k}=diag\left(L_{k,1},L_{k,2},\ldots ,L_{k,m}\right)$, $\Psi_{k}=\left[\Psi_{k,1}^{T},\Psi_{k,2}^{T},\ldots ,\Psi_{k,m}^{T}\right]$.

## 5. Simulations

In this section, the performance of the proposed filter is illustrated by a state estimation in the linear system. Matlab 2018b was used in the simulation on the computer with Intel(R) Core(TM) i5-1035G1 CPU @ 1.00 GHz 1.19 GHz. The dynamical equation of the system is given by
(12)$$x_{k+1}=\left[\begin{array}{cc} 0.9996 & -0.0300\\ 0.0300 & 0.9996 \end{array}\right]x_{k}+\left[\begin{array}{cc} 0.015 & 0\\ 0 & 0.015 \end{array}\right]w_{k}$$
where $x_{k}$ is the state of the stem at the *k* instance, $w_{k}$ is a discrete random process with zero means, and its covariance matrix is Q=diag([2,2]). The initial value of the system state is $x_{0}={\left[7.5,-5\right]}^{T}$.

Here, we consider a WSN composed of m=100 sensor nodes located in a square region with a 1000 m side length. The topologies of WSNs under two fixed communication radii are shown in Figure 4, where the communication radius of the left figure is $d_{min}=160$ m and another is $d_{max}=260$ m. If two nodes are connected, it means that they are able to receive local information from each other, otherwise, they are not. It is easy to see from Figure 4 that the complexity degree of WSNs is completely determined by the communication radius.

The detection value provided by each sensor node can be defined as
(13)$$y_{k,i}=\left[\begin{array}{cc} 1 & 0\\ 0 & 1 \end{array}\right]x_{k}+v_{k,i}$$
where $v_{k,i}$ is the measurement noise with zero means and its covariance matrix is $R_{i}=diag\left(\left[rand,rand\right]\right)$. The rand denotes uniformly distributed random numbers in (0,1). In addition, let c=0.001, as well as the initial energy of each node be 2 J. Other parameters in the simulation are set in Table 3.

In order to show the performance of the filter, it is expressed in terms of the root mean square error (RMSE): (14)$${\mathrm{RMSE}}_{k}=\sqrt{\frac{1}{m}\sum_{i=1}^{m}{\left(x_{k}-x_{k,i}^{+}\right)}^{T}\left(x_{k}-x_{k,i}^{+}\right)}$$

For showing the filter performance proposed in this paper, the different communication patterns are compared. Pattern 1: our algorithm (using the event-triggered schedule and topology transformation schedule, i.e., $d=d_{max}$ or $d=d_{min}$). Pattern 2: distributed Kalman filter (using the fixed large communication radius, i.e., $d=d_{max}$). Pattern 3: distributed Kalman filter (using the fixed small communication radius, i.e., $d=d_{min}$). The RMSEs of three patterns are depicted in Figure 5.

Overall, their filtering accuracy is comparable. The performance of pattern 1 is better after the 140th step. This trend is much more obvious as time goes by. Compared with pattern 1, the node that first runs out of energy (i.e., dead node) appears earlier in other patterns (see Figure 6), which leads to the deterioration of the topology connectivity. Thus, there is a decrease in the filtering accuracy of patterns 2 and 3 between k=140 and k=200.

Figure 6 shows the energy consumption change of the node that first runs out of energy in the three patterns. It is evident that the dead node in pattern 1 appears later (around the 200th step), indicating that it can significantly extend the lifetime of WSNs. Specifically, compared to pattern 3, it prolongs the lifetime by about 40%, let alone pattern 2. This shows that the proposed topology transformation strategy is highly effective in energy conservation.

However, if the parameters are not selected suitably, the above result cannot be obtained. For example, let δ=0.25, the lifetimes of WSNs in pattern 1 and pattern 3 will be changed (see Figure 7).

To further illustrate the effectiveness of the method, the event-triggered frequency and communication distance of the nodes in WSNs at every time *k* are shown in Figure 8 and Figure 9, respectively (in order to clearly display the figure, the event-triggered numbers at time k=0 of the three patterns are deleted).

In Figure 8, the total event-triggered frequency (TEF) in three patterns is 529 freq, 377 freq, and 882 freq, respectively, which means that the event-triggered frequency of our algorithm is medium. Compared to pattern 1 (our algorithm), the event-triggered frequency in pattern 2 is reduced by 36.30% and increased by 66.73%, respectively. Thus, our algorithm is more effective in reducing the event-triggered frequency. This suggests that the event-triggered condition proposed in this paper is helpful for energy saving.

In addition, in Figure 9, the communication radius of the nodes is switched by our algorithm, which evidences the effectiveness of the proposed topology transformation schedule. According to the proposed topology transformation schedule (see Equation (Equation 7), if the energy consumption of node *i* is lower than the local average energy consumption, the communication radius switches to the $d_{min}$ at the next time. Otherwise, it switches to the $d_{max}$. Therefore, it can make more uniform energy consumption (see Figure 6 and Figure 7; the absolute value of the slope of the curve in pattern 1 is the smallest in the three patterns). These results further prove the effectiveness of the proposed algorithm.

In order to show the effectiveness of different parameters on the performance of WSNs, we conducted the experiment using the statistics method. The results are shown in Table 4, Table 5 and Table 6.

From Table 4, it can be observed that the total frequency increases as the packet loss rate α increases, resulting in an increase in filtering accuracy but a decrease in the lifetime of WSNs. Additionally, Table 5 indicates that the total frequency decreases as the event-triggered threshold δ increases, resulting in a decrease in filtering accuracy but an increase in the lifetime of WSNs. Table 6 shows that the timeliness window Δ contributes to improved filtering accuracy and the lifetime of WSNs, but larger values of Δ have a negative impact on them. Thus, we need to adjust the parameters to obtain the desired filtering accuracy and the expected lifetime WSNs.

## 6. Conclusions

In this paper, based on the timeliness window, an energy-saving distributed consensus Kalman filter with a dual event-driven strategies was designed for WSNs. It is a comprehensive algorithm for saving energy and for uniform energy consumption. On the one hand, the proposed event-triggered schedule based on the timeliness window saves energy, satisfying the filtering accuracy. On the other hand, the topological transformation schedule, which chiefly controls the topology structure, was designed according to the energy consumption model. To be more specific, it is able to switch the communication radius according to the proposed topology transformation schedule, which makes the energy consumption uniform. The following are the highlights of this paper:The unique dual event-driven strategy was designed to balance the filtering accuracy and the energy consumption. Using the proposed dual event-driven strategy, the lifetime of WSNs can be extended by about 40%.A novel distributed consensus Kalman filter was designed based on the two schedules; sufficient conditions for the stability of the filter are given.

Simulation tests have demonstrated the effectiveness of the proposed event-triggered schedule and topology transformation schedule in achieving a better trade-off between estimated accuracy and energy-saving by adjusting various parameters, ultimately leading to a prolonged lifetime of WSNs. However, there is a problem in that the proposed algorithm depends on the choice of parameters. The different parameters can lead to significant changes in the performance of the proposed algorithm. It is widely known that the intelligent optimization algorithm can be used to adjust the parameters to obtain the desired performance indicators. Thus, it is natural to expect that it will be solved by the intelligent optimization algorithm in future works.

## Figures and Tables

**Figure 1 sensors-23-03261-f001:**
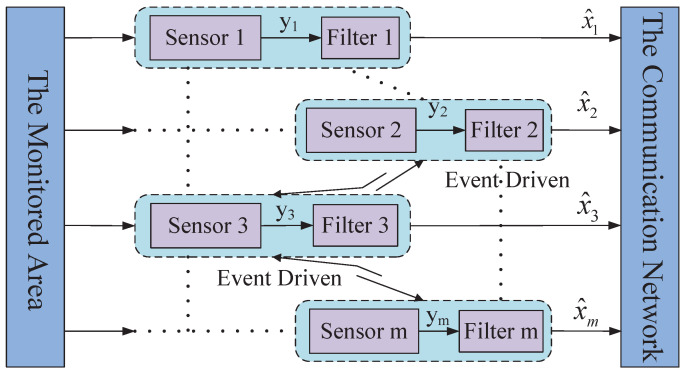
Frame of the event-driven distributed filter in the monitored area.

**Figure 2 sensors-23-03261-f002:**
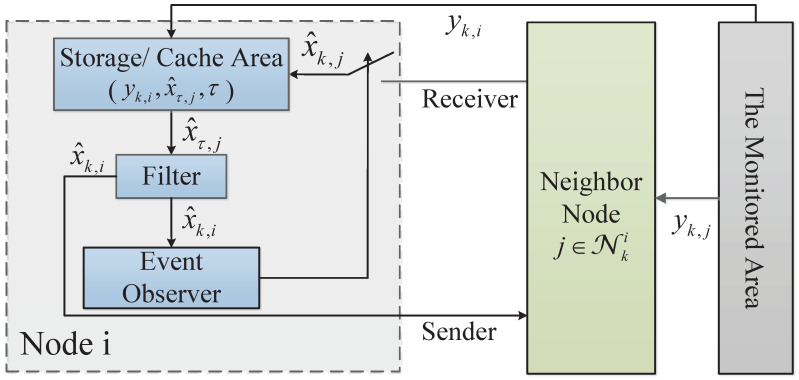
Diagram of the event-triggered principle.

**Figure 3 sensors-23-03261-f003:**
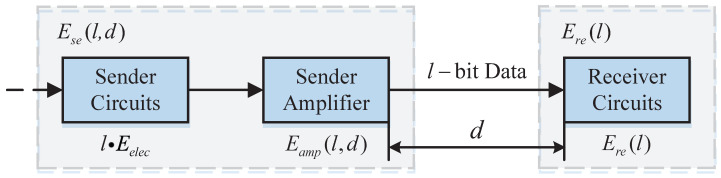
The energy consumption model of nodes in WSNs.

**Figure 4 sensors-23-03261-f004:**
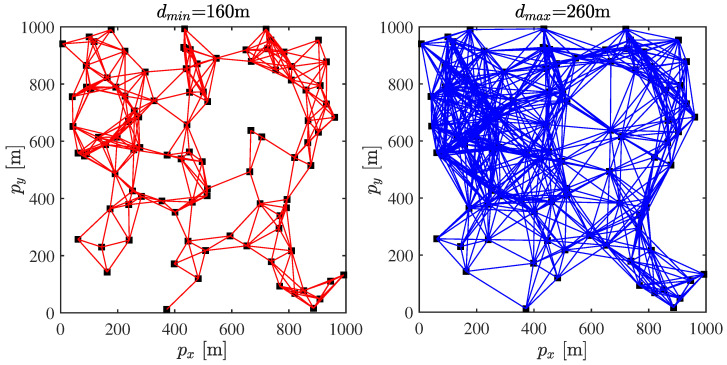
The topologies of WSNs under two fixed communication radii.

**Figure 5 sensors-23-03261-f005:**
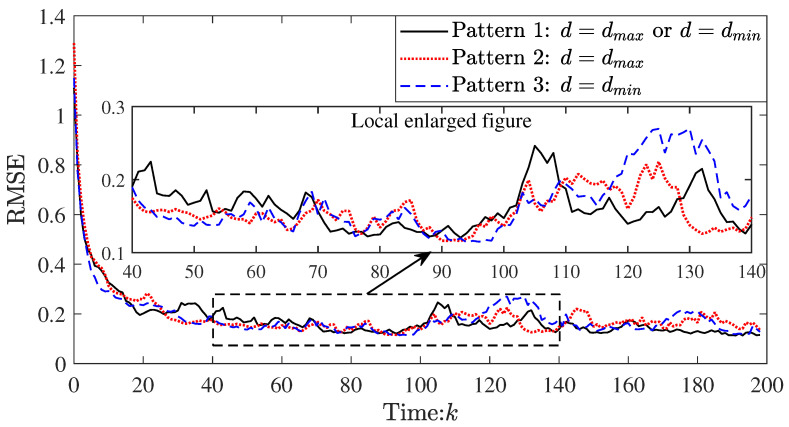
The RMSEs of three patterns.

**Figure 6 sensors-23-03261-f006:**
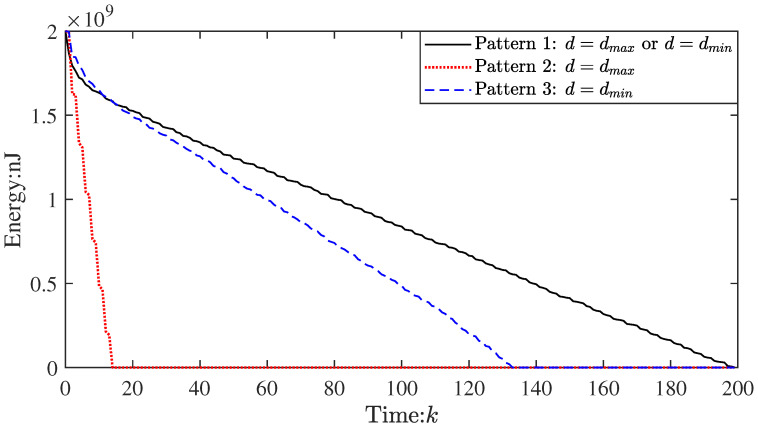
The energy consumption change of the node that first runs out of energy in the three patterns when δ=0.8.

**Figure 7 sensors-23-03261-f007:**
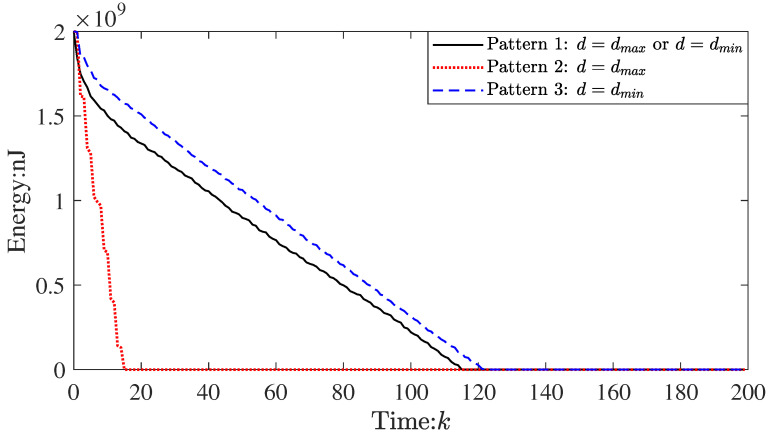
The energy consumption change of the node that first runs out of energy in three patterns when δ=0.25.

**Figure 8 sensors-23-03261-f008:**
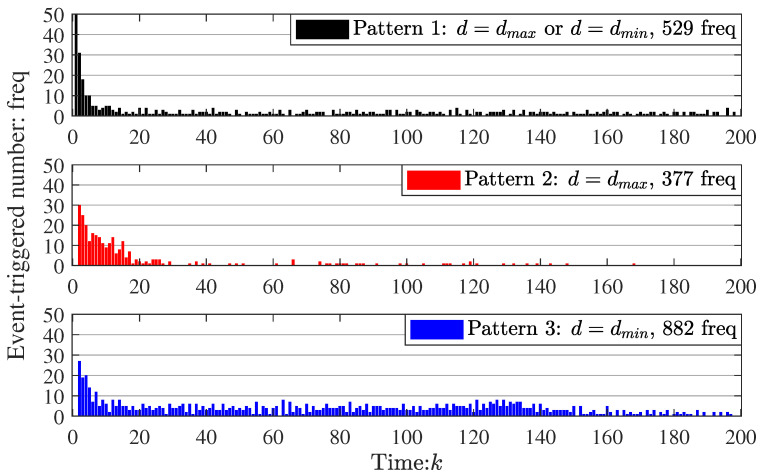
The event-triggered frequency of the node at every time *k*.

**Figure 9 sensors-23-03261-f009:**
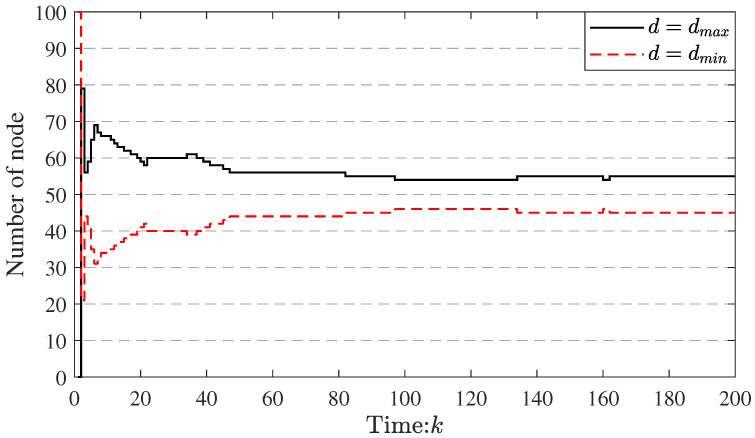
The number of nodes used at different communication distances at every time *k*.

**Table 1 sensors-23-03261-t001:** List of important notations.

Symbol	Definition
$x_{k}\in R^{n\times 1}$	The state of the monitored object at the *k* instance
${\hat{x}}_{k,i}^{-}$	The prior estimate of the system state for node *i* at the *k* instance
${\hat{x}}_{k,i}^{+}$	The posterior estimate of the system state for node *i* at the *k* instance
$P_{k,i}^{-}$	The covariance matrix of the prior estimate error for node *i* at the *k* instance
$P_{k,i}^{+}$	The covariance matrix of the posterior estimate for node *i* at the *k* instance
$w_{k}$	The system noise at the *k* instance
$y_{k,i}$	The output of the *i*-th sensor node at the *k* instance
$v_{k,i}$	The measurement noise of the *i*-th sensor at the *k* instance
*Q*	The covariance matrix of the system noise
$R_{k,i}$	The covariance matrix measurement noise of the *i*-th sensor at the *k* instance
$K_{k,i}$	The Kalman gain of the *i*-th sensor at the *k* instance
$C_{k,i}$	The consensus gain of the *i*-th sensor at the *k* instance
$\Delta \in Z^{+}$	The timeliness period
$N_{TW}^{i}$	The set of timeliness neighbors of node *i*
$N_{RN}^{i}$	The set of real-time neighbors of node *i*
$N_{EN}^{i}$	The set of effective neighbors of node *i*
δ	The event-triggered threshold
$\tau_{k}^{i}\in \left\{0,1,2,\ldots ,\Delta \right\}$	The difference between the last event-triggered time of node *i* and the current
$E_{se}$	The energy consumed by the sender
$E_{re}$	The energy consumed by the receiver
$E_{i}^{to}$	The total energy consumption of node *i* in a data fusion process
$E_{k,i}^{av}$	The local average energy of node *i* at the *k* instance
$d_{k,i}$	The communication radius of node *i* at the *k* instance
α	The packet loss rate in WSNs

**Table 2 sensors-23-03261-t002:** List of abbreviations.

Acronym	Definition
WSNs	Wireless sensor networks
TW	Timeliness window
RN	Real-time neighbor
EN	Effective neighbor
RMSE	Root mean square error
TEF	Total event-triggered frequency

**Table 3 sensors-23-03261-t003:** The values of other parameters in the simulation.

Parameters	δ	Δ	α	*l*	$d_{0}$
Value	0.8	5	0.3	40,000 bit	200 m
$x_{0,i}^{-}$	$P_{0,i}^{-}$	$E_{elec}$	$E_{ce}$	$\varepsilon_{fs}$	$\varepsilon_{mp}$
${\left[0,0\right]}^{T}$	diag([5,5])	50 nJ/bit	5 nJ/(bit·signal)	10 pJ/(bit$\cdot m^{2}$)	0.0013 pJ/(bit$\cdot m^{4}$)

**Table 4 sensors-23-03261-t004:** The effects of different α values on the performance of WSNs, when δ=0.8 and Δ=5.

	Parameter	α=0.2	α=0.4	α=0.6
Performance				
RMSE		0.2003	0.1982	0.1968
Lifetime		198	190	185
TEF		522	581	604

**Table 5 sensors-23-03261-t005:** The effect of different δ values on the performance of WSNs, when α=0.3 and Δ=5.

	Parameter	δ=0.4	δ=0.6	δ=0.8
Performance				
RMSE		0.1861	0.2034	0.2133
Lifetime		182	193	200
TEF		746	654	514

**Table 6 sensors-23-03261-t006:** The effect of different δ values on the performance of WSNs, when α=0.3 and δ=0.8.

	Parameter	Δ=1	Δ=5	Δ=10
Performance				
RMSE		0.1925	0.1836	0.2365
Lifetime		186	200	167
TEF		584	521	659

## Data Availability

All data generated or analyzed during the study are included in this published article.

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
