# Peer review of "Distributed Consensus Kalman Filter Design with Dual Energy-Saving Strategy: Event-Triggered Schedule and Topological Transformation"

_sensors, 2023, doi:10.3390/s23063261_

Round 1

Reviewer 1 Report

This study describes in relatively detail a timelines window-based energy saving strategy for WSNs.

However, some supplements are needed for subscribers.

1. Overall, are the formulas or developments in the text invented by the author? If not, you can give appropriate references.

2. The tool and SW used in the simulation method must be explained indispensably. The simulation results look relatively reviewable, but I don't understand the method at all.

3. The explanation for Fig 5 and the interpretation of the results are unknown. The author must be specifically described and can be deleted if necessary. Why should state response and trajectory curves be analyzed?

4. Why is the communication distance fixed at 160 and 260m in the simulation? In addition, the definition of communication distance should be clearer.

5. line 258, "However, if parameters are not selected suitably, the above result can not be obtained" This sentence confuses the author's findings. It is necessary to clearly state why figs 7 and 8 are different.

6. Overall, the results of the study are relatively inconsistent. (Especially, figs 7 and 8) Also, for the author's research to be valuable, a lot of thought and discussion is needed in the conclusion part.

Author Response

Details of the response can be seen in attachement.

Reviewer 2 Report

In this article useful information on Distributed Consensus Kalman Filter Design with Dual Energy Saving Strategy: Event-triggered Schedule and Topological Transformation has been provided. However, author need to address following comments in order to publish in this journal.

1.      Introduction is very general without any data. Author should add overall energy consumption and emission data . They can refer following most related articles for related information: https://doi.org/10.1016/j.desal.2017.03.009

2.      Equations need more explanation.

3.      Many statements in the introduction are without references. Authors should reference each statement from where it was extracted.

4.      Overall English need to improve.

Article need minor revision.

Author Response

Details of the response can be seen in attachment.

Reviewer 3 Report

This paper designs an energy-saving distributed consensus Kalman filter with dual event-driven strategies for WSNs to try to save energy after satisfying the filtering accuracy. The results show that the algorithm can better trade off the relationship between the estimated accuracy and the energy saving by setting different parameters. However, it still has the following problems:

1.     The connection of the introduction is not natural, please optimize it.

2.     Please summarize the contribution of the paper and take it as part of the introduction.

3.     The proof in the paper is excessive, which reduces the readability of the content. Please delete it appropriately. A list of abbreviations and symbols is also needed.

4.     The performance of the proposed method cannot be compared with the most advanced results. Please add relevant comparative experiments.

5.     The summary of the results is too general. Please elaborate on the results, which are actually more intuitive.

Author Response

(The authors gave the same response as above.)

Reviewer 4 Report

This paper designed an energy-saving distributed consensus Kalman filter with dual event-driven strategies based on the timeliness window for WSNs. The idea is clear and the paper is organized well. But the paper still has some problems. It is recommended to modify.

1.       I found some writing errors in the paper. For example, “then” in “then, it can be given that” in 202 linear should be changed to “Then”. Please proof-read the manuscript against the template carefully to minimize typographical, grammatical and bibliographical errors.

2.    The captions of Figures 7 and 8 are the same. It is suggested to modify the captions of these two figures.

3.    In the analysis of the results, there is a conclusion that the proposed method can improve the accuracy of the filtering. But I cannot draw this conclusion from Figure 6. There does not seem to be a clear trend in the results of Figure 6. It is recommended that more data and analysis be added to demonstrate this.

4.       The results of the different patterns is not clear in Figures 9-10. It is suggested to modify these two figures to make the comparison of different patterns clearer.

Author Response

(The authors gave the same response as above.)

Round 2

Reviewer 1 Report

Some doubts have been cleared.

1. Fig. 8 and 9 need zooming. In particular, fig8 needs to be clearly expressed by adjusting the y-axis.

2. Conclusions are areas that highlight the author's research.

 The considerations in this study and the limitations of parameter-dependent algorithms need to be further described.

 And it is worth mentioning specific future research that could overcome the problem.

Author Response

Many thanks for your nice suggestions. We have improved our manuscript according to your advices. Details could be seen in attachement.
